# Green Synthesized Gold and Silver Nanoparticles Increased Oxidative Stress and Induced Cell Death in Colorectal Adenocarcinoma Cells

**DOI:** 10.3390/nano13071251

**Published:** 2023-04-01

**Authors:** Cristina Bidian, Gabriela Adriana Filip, Luminița David, Bianca Moldovan, Diana Olteanu, Simona Clichici, Maria-Cristina Olănescu-Vaida-Voevod, Cristian Leostean, Sergiu Macavei, Dana Maria Muntean, Mihai Cenariu, Adriana Albu, Ioana Baldea

**Affiliations:** 1Department of Physiology, ‘‘Iuliu Hatieganu’’ University of Medicine and Pharmacy, 1-3 Clinicilor Street, 400006 Cluj-Napoca, Romania; cristina.bidian@umfcluj.ro (C.B.); olteanu_diana_elena@yahoo.com (D.O.); sclichici@umfcluj.ro (S.C.); cristina_vaida2005@elearn.umfcluj.ro (M.-C.O.-V.-V.); ioana.baldea@umfcluj.ro (I.B.); 2Department of Chemistry, Faculty of Chemistry and Chemical Engineering, “Babes-Bolyai” University, 11 Arany Janos Street, 400028 Cluj-Napoca, Romania; bianca.moldovan@ubbcluj.ro; 3National Institute for Research and Development of Isotopic and Molecular Technologies, Donath St., No. 67-103, 400293 Cluj-Napoca, Romania; cristian.leostean@itim-cj.ro (C.L.); sergiu.macavei@itim-cj.ro (S.M.); 4Department of Pharmaceutical Technology and Biopharmaceutics, ‘‘Iuliu Hatieganu’’ University of Medicine and Pharmacy, 8 Victor Babeș Street, 400347 Cluj-Napoca, Romania; dana.muntean@umfcluj.ro; 5Department of Animal Reproduction, University of Agricultural Sciences and Veterinary Medicine, 3-5 Calea Manastur Street, 400372 Cluj-Napoca, Romania; mihai.cenariu@usamvcluj.ro; 62nd Department of Internal Medicine, “Iuliu Hatieganu” University of Medicine and Pharmacy, 8 Babes Street, 400012 Cluj-Napoca, Romania; adriana.albu@umfcluj.ro

**Keywords:** gold and silver nanoparticles, human colorectal adenocarcinoma cells, apoptosis, necrosis, oxidative stress

## Abstract

The research investigated the effect of gold (Au-CM) and silver nanoparticles (Ag-CM) phytoreduced with *Cornus mas* fruit extract (CM) on a human colorectal adenocarcinoma (DLD-1) cell line. The impact of nanoparticles on the viability of DLD-1 tumor cells and normal cells was evaluated. Oxidative stress and cell death mechanisms (annexin/propidium iodide analysis, caspase-3 and caspase-8 levels, p53, BCL-2, BAX, NFkB expressions) as well as proliferation markers (Ki-67, PCNA and MAPK) were evaluated in tumor cells. The nanoparticles were characterized using UV–Vis spectroscopy and transmission electron microscopy (TEM) and by measuring zeta potential, hydrodynamic diameter and polydispersity index (PDI). Energy dispersive X-ray (EDX) and X-ray powder diffraction (XRD) analyses were also performed. The nanoparticles induced apoptosis and necrosis of DLD-1 cells and reduced cell proliferation, especially Ag-CM, while on normal cells, both nanoparticles maintained their viability up to 80%. Ag-CM and Au-CM increased the expressions of p53 and NFkB in parallel with the downregulation of BCL-2 protein and induced the activation of caspase-8, suggesting the involvement of apoptosis in cell death. Lipid peroxidation triggered by Ag-CM was correlated with tumor cell necrosis rate. Both nanoparticles obtained with phytocompounds from the CM extract protected normal cells and induced the death of DLD-1 tumor cells, especially by apoptosis.

## 1. Introduction

Oncological diseases are a significant cause of morbidity and mortality worldwide, GLOBOCAN 2020 indicating that in 2020, globally, there were 19.3 million new cases and almost 10 million deaths due to cancer. Of the new cases, 10% are represented by colorectal cancer [1]. Among the most common risk factors are the patient’s age of over 50 years, family or personal history of polyps or colorectal cancer, inflammatory bowel diseases, low-fiber diet and sedentary lifestyle. The standard treatment is surgery combined with chemotherapy and/or radiotherapy, sometimes with a high recurrence rate. In selected cases, targeted biological therapy or immunotherapy are of great benefit. 

The use of nanomaterials has emerged as a promising additional method in the treatment of colorectal cancer, with the aim of improving therapeutic results and reducing the side effects of conventional therapies.

Gold nanoparticles (AuNPs) and silver nanoparticles (AgNPs) are used as antitumor agents because they have specificity for tumor cells, where they tend to accumulate [2,3,4]. Nanoparticles can act passively, as nanocarriers, or actively, through ligand–receptor interaction [5] and induce DNA damage with cell cycle arrest, mitochondrial changes, aberrant regulation of p53 protein, caspase-3 activation and, consequently, apoptosis [6,7,8].

AuNPs are used as drug carriers, molecular probes or biosensors [9]. Their access to the tumor microenvironment is conditioned by biocompatibility, small size, optical properties, as well as the possibility of easy modification of their surface [10,11].

AgNPs release silver ions and produce intracellular free radicals, with disruption of different intracellular mechanisms, leading to cell damage and cell death [12]. AgNPs enter cells by endocytosis and are transferred to lysosomes. The acidic lysosomal environment degrades the nanoparticles with the release of Ag^+^ ions into the cytosol [13]. Once internalized, nanoparticles release high levels of toxic ions (“Trojan horse” mechanism) [14]. In addition, Ag ions bind to thiol groups in enzyme structures and form stable complexes, altering enzyme configuration and activity [15,16,17,18]. Furthermore, in low concentration, AgNPs can trigger DNA damage and chromosomal alterations without significant systemic toxicity [19,20].

Green biosynthesis of metallic nanoparticles is an environmentally friendly and inexpensive process that uses microorganisms or plants as source of reducing agents. In addition, plant extracts have antioxidant, anti-inflammatory, antibacterial, antiviral and antifungal properties and can be used for nanoparticle surface conversion. Cornelian cherry fruits are remarkable sources of bioactive molecules which can successfully act as reducing agents for the noble metals ions in order to obtain metallic nanoparticles as well as stabilizing and capping agents of the synthesized nanostructures. The CM fruits are reported to contain high amounts of antioxidant compounds such as anthocyanins, flavonoids and phenolic acids. The main components of the cornelian cherry fruit extract, as identified in one of our previous studies, [21] are pelargonidin-3-O-glucoside, cyanidin 3-O-glucoside, cyanidin-3-O-galactoside, pelargonidin-3-O-rutinoside, kaempferol-3-O-galactoside, quercetin-3-O-glucuronide, chlorogenic acid, caffeic acid, ellagic acid catechin and epicatechin. All these compounds are responsible for the wide range of health-promoting effects of the CM fruit extract and can enhance the biological activities of gold and silver nanoparticles by their surface modification.

Phytosynthesized nanoparticles, after binding to the cell surface or internalization [22], can damage the cell membrane and lead to disruption of transmembrane electron transport, generate reactive oxygen species (ROS) and trigger cell apoptosis [23,24,25]. Moreover, phytochemical compounds can promote the internalization of nanoparticles in tumor cells, increase oxidative stress [26,27,28] and DNA damage [29], initiate alteration of the electron transport chain [23], enzyme or protein oxidation [30,31], leading to disruption of their activity and apoptosis.

In addition, nanoparticles are attracted to tumor cells by electrostatic forces [32] and can dissociate into ions, which penetrate the cells, thus exerting antitumor effects [33].

Based on these properties, our research investigated the antiproliferative effect of green synthesized gold (Au-CM) and silver (Ag-CM) nanoparticles with *Cornus mas* extract, on a human colorectal adenocarcinoma (DLD-1). It is known that these fruits have an increased content of flavonoids, anthocyanins, phenolic acids with reducing properties and antioxidant, anti-inflammatory and anticancer effects [34,35]. The impact of nanoparticles on dermal fibroblasts and tumor cell viability as well as oxidative stress and cell death mechanisms on tumor cells were studied. UV–Vis spectroscopy, transmission electron microscopy (TEM), determination of zeta potential and hydrodynamic diameter were used to characterize the nanoparticles. Energy dispersive X-ray (EDX) and X-ray diffraction (XRD) analyses were also performed.

## 2. Materials and Methods

### 2.1. Reagents

Folin–Ciocalteu reagent, silver nitrate and gold chloride solution, as well as Bradford reagent and 2-thiobarbituric acid were purchased from Sigma-Aldrich Chemicals GmbH (Darmstadt, Germany). Kits for capase-3 and capase-8 determinations were purchased from Elabscience (Houston, TX, USA). Antibodies against p53, BCL2, BAX, NFkB, pNFkB proteins and GAPDH were purchased from Santa Cruz Biotechnology (Santa Cruz, CA, USA). Antibodies against Ki-67 were purchased from Invitrogen (Rockford, IL, USA) and those against PCNA and MAPK were purchased from Cell Signaling Technology (Danvers, MA, USA). Propidium iodide and Annexin V-fluorescein isothiocyanate were purchased from BD Pharmingen Biosciences (San Jose, CA, USA). The CellTiter 96^®^ cell proliferation assay was purchased from Promega Corporation (Madison, WI, USA). All reagents used were of high purity.

### 2.2. Obtaining Cornelian Cherry Fruit Extract and Its Characterization of Fruit Extract

The Cornelian cherries were purchased from a local market from Cluj-Napoca in July 2021 and kept frozen until use. For the preparation of the extract, the fruit flesh puree was mixed with distilled water (in ratio of 1:20). The obtained mixture was stirred for an hour at 22 °C, and then filtered at low pressure. The resulted solution was used to obtain the metallic nanoparticles. Total phenolic content (TPC), expressed as µg galic acid equivalents (GAE)/mL extract, was determined by the Folin–Ciocalteu assay [36], slightly modified and previously described [37], and was used as key parameter for the characterization of Cornelian cherry extract.

### 2.3. Preparation and Characterization of Nanoparticles

The silver and gold nanoparticles were obtained by reducing the metallic ions, using the bioactive compounds from Cornelian cherry extract as reducing and stabilizing molecules according to the following procedures:a.In order to obtain the silver nanoparticles, 10 mL fruit extract was added to 30 mL aqueous silver nitrate 1 mM solution, under stirring, at 22 °C. The pH of the resulted mixture was fixed at 8 by adding NaOH solution (0.1 M). After 30 min, the light red color of the solution changed to brown-yellow, indicating the formation of colloidal silver.b.In order to prepare gold nanoparticles, 10 mL of fruit extract (brought to pH = 7.5 with 0.1 M NaOH solution) was added, under stirring, to 40 mL boiling solution of 1 mM tetrachloroauric acid. A red-purple solution of colloidal gold was obtained after 30 min of stirring

The silver and gold nanoparticles thus obtained were characterized by means of consecrated analytical methods, namely UV–Vis spectroscopy (using a Perkin Lambda 25 spectrophotometer), transmission electron microscopy (TEM—using a Hitachi H-7650 120 kV automatic transmission electron microscope), measurement of hydrodynamic diameter and zeta potential (Zetasizer Nano ZS-90 instrument, Malvern Instruments Ltd., Malvern, UK). Determinations were performed in triplicate and the mean value was reported.

X-ray diffraction (XRD) data were acquired with a Smart Lab Rigaku diffractometer with a graphite monochromator with Cu-Kα radiation (1:54 Å) X-ray source: Anode Cu, 9 kW at room temperature over the 2Theta range from 10 to 90 degrees, with a 0.01-degree step. For the Rietveld refinement, the Integrated X-ray Powder Diffraction (PDXL) software was used. The morphology of both nanoparticles was investigated by Scanning Transmission Electron Microscopy (STEM) type HITACHI HD-2700, cold field emission electron beam accelerated at 200 kV and 10 μA, EDX and diffraction analyses, magnification 100×–10,000,000×.

### 2.4. Biological Assays

#### 2.4.1. Cell Cultures

Human colorectal adenocarcinoma cells (ECACC 90102540) (passage 31) and human dermal fibroblasts (HDF) were purchased from Sigma Aldrich Chemie GmbH (Darmstadt Germany). Passages 32–33 in adenocarcinoma cells and passage 2 in fibroblasts were used. The 2 cell lines were cultured in Dulbecco’s modified Eagle’s medium (DMEM), supplemented with fetal bovine serum (5%), hydrocortisone (5 μg/mL for DLD-1), antibiotics and antifungals.

#### 2.4.2. Cell Viability Assay

Cell viability testing was determined with the CellTiter 96^®^ Aqueous Non-Radioactive Cell Proliferation assay (Promega Corporation, Madison, WI, USA) according to the manufacturer’s instructions. Formazan is a colored compound synthesized by viable cells and was determined colorimetrically. The 2 cell lines were seeded in 96-well plates (purchased from TPP, Trasadingen, Switzerland) for 24 h at a density of 10^4^/wells and then exposed to silver nanoparticles (Ag-CM) and gold nanoparticles (Au-CM) (0–100 μg/mL silver/gold content) and CM extract suspended in medium for 24 h (0–100 μg polyphenols/mL). Cell viability was measured spectrophotometrically using an ELISA plate reader (purchased from Tecan, Switzerland) at a wavelength of 540 nm. All experiments were conducted in triplicate. They were compared with the values of untreated cell cultures (control cells). The dose at which cell viability fell below 70% was considered toxic. 

#### 2.4.3. Cell Lysates

DLD-1 cells seeded on Petri dishes (density 10^4^/cm^2^) were exposed for 24 h to CM extract (40 μg polyphenols/mL), Ag-CM (4 μg/mL) or Au-CM (40 μg/mL). Untreated cells represented the controls. After exposure, cells were washed, and then lysates were prepared according to the method described by Baldea el al. (2015) [38]. Protein concentrations were determined by the method specified by the manufacturer (Biorad, Hercules, CA, USA). Bovine serum albumin was used as a standard. Finally, cell lysates were corrected for total protein concentration. Cell lysates were used for MDA and caspase assessment.

#### 2.4.4. Oxidative Stress Assessment

Malondialdehyde (MDA), a marker of oxidative stress, was assessed by spectrophotometry following a slightly modified Conti method [39]. Results were expressed in nmol/mg protein.

#### 2.4.5. Cell Death Mechanisms and Proliferation Markers

The cell death was assessed by flow cytometry, and the mechanisms involved in cell death were quantified by Western blotting and caspase-3 and caspase-8 estimation. For annexin/propidium iodide analysis, tumor cells (DLD-1) were seeded in Petri dishes (at a density of 10^4^/cm^2^), then exposed for 24 h to CM extract (40 μg polyphenols/mL), Ag-CM (4 μg/mL) or to Au-CM (40 μg/mL). Unexposed cells were controls. After exposure, cells were washed and stained with annexin V/propidium iodide (PI) purchased from BD Pharmingen Biosciences (San Jose, CA, USA), then analyzed by flow cytometry according to the method described by Moldovan et al. 2012 [37]. Results were reported as a percentage of the total number of cells and interpreted as follows: viable cells—annexin V negative and PI negative cells; early apoptosis—annexin V positive and PI negative cells; late apoptosis—annexin V positive and PI positive cells; necrosis—annexin V negative and PI positive cells.

Transcription factors, apoptosis regulators and proliferation markers were evaluated by Western blotting. Cell lysates (containing 20 μg protein/lane) were prepared for Western blotting as follows: they were separated by gel electrophoresis and then transferred to polyvinylidene difluoride membranes. For this, the Biorad Miniprotean system (purchased from Biorad, Hercules, CA, USA) was used. Blots were subsequently blocked and incubated with antibodies against p53, BCL-2, BAX, NFκB, pNFκB proteins. Finally, the blots were washed and incubated with peroxidase-linked antibodies (purchased from Santa Cruz Biotechnology, Santa Cruz, CA, USA). Proteins were highlighted using a Supersignal West Femto Chemiluminescent substrate (purchased from Thermo Fisher Scientific, Hanover Park, IL, USA) and an imaging system equipped with an XRS camera and analysis software (purchased from Biorad, Hercules, CA, USA). Protein loading control was performed with GAPDH (purchased from Trevigen Biotechnology, Gaithersburg, MD, USA). Caspase-3 and caspase-8 levels were assessed in cell lysates using ELISA kits, following the manufacturer’s recommendations. Results were expressed in pg/mg protein.

### 2.5. Data Analysis

Statistical processing was performed with GraphPad Prism version 5.0 for Windows, GraphPad Software (San Diego, CA, USA). One-way ANOVA and the Bonferroni test were used. The results were interpreted as statistically significant at *p* < 0.05 (* *p* < 0.05, ** *p* < 0.01, *** *p* < 0.001).

## 3. Results

### 3.1. Characterization of Metallic Nanoparticles

The reduction in silver ions (from AgNO_3_ solution) and gold ions (from HAuCl_4_ solution) to form the corresponding metallic nanoparticles was monitored by UV–Vis spectroscopy. The UV–Vis spectra of Cornelian cherries extract and the obtained silver and gold nanoparticles are presented in Figure 1. As one can see, the UV–Vis spectrum of Cornelian cherry fruit extract exhibits a peak at λ_max_ = 508 nm, characteristic for the anthocyanins present in the solution. The UV–Vis spectrum of green synthesized silver nanoparticles shows a maximum at 405 nm, which represents a characteristic value for the Plasmon resonance of metallic silver [40], while in the spectrum of AuNPs, the peak located at 527 nm confirms the formation of colloidal metallic gold [41].

The shape and morphology of the biosynthesized metallic nanoparticles were investigated using transmission electron microscopy (TEM). The analysis of TEM images (Figure 2) demonstrates the formation of almost spherically shaped metallic nanoparticles, with an average diameter of 17 nm for Ag-CM (Figure 2a) and 20 nm for Au-CM (Figure 2b).

Zeta potential, measured by laser Doppler microelectrophoresis, was −31 mV for Ag-CM and −24.4 mV for Au-CM (Figure 3).

The hydrodynamic diameter of the nanoparticles after binding the polyphenols from the CM extract was 326 nm for Ag-CM and 229 nm for Au-CM, respectively (Figure 4).

The XRD analysis confirmed that the sample is silver; the identification ICDD DB card number is 00-002-1098. Three distinctive diffraction peaks at 2θ values (2theta(deg)38.277), 44.14(3), and 77.65(8)) that corresponded to the reflection planes of (111), (200), and (311), characteristic of Space group 225: Fm-3m, silver were observed. The crystalline size, as determined through the Williamson–Hall method, was of 10.77(3) Å, indicating the successful formation of a silver nanoparticle (Figure 5).

The XRD analysis confirmed that the sample is gold; the identification ICDD DB card number is 00-001-1172. Three distinctive diffraction peaks at 2θ values (2-theta(deg)56.95(17), 64.6(6), and 77.4(4)) that corresponded to the reflection planes of (111), (200), (220) characteristic of gold were observed. The crystalline size, as determined through the Williamson–Hall method, was of 5.8 Å, indicating the successful formation of a gold nanoparticle (Figure 5).

According to the elemental distribution as shown by the EDX analysis, we confirmed the presence of silver and gold nanoparticle; spectra were obtained using Oxford Instrument windowless detector and AZtec Software. The samples were prepared by dropping a few μL of diluted ethanol suspension of the sample on the copper grid (Figure 6).

### 3.2. Biological Assays

#### 3.2.1. Cell Viability Assay

The MTS assay evaluated the effects of Au-CM and Ag-CM nanoparticles on HDF and DLD-1 cell viability compared to untreated normal cells and cells exposed to CM extract. Thus, CM extract did not significantly alter the viability of HDF cells up to concentrations of 50 μg/mL, whereas at concentrations higher than 75 μg/mL, the viability significantly decreased in a dose-dependent manner. Exposure of HDF cells to Au-CM decreased the viability starting at 50 μg/mL and reached a minimum at 75 μg/mL while higher concentrations did not further affect the cell viability. Exposure of HDF cells to Ag-CM decreased cell viability from a concentration of 10 μg/mL, progressive decrease being further registered with increasing concentration of Ag-CM. The highest toxicity of Ag-CM to HDF cells was observed at 100 μg/mL (Figure 7).

Treatment of DLD-1 cells with CM extract significantly decreased cell viability starting at concentrations of 50 μg/mL, while at 75 μg/mL and 100 μg/mL, cell viability remained unaffected. Au-CM exerted toxicity on DLD-1 starting at 50 μg/mL, with cell viability progressively decreasing in correlation with increasing Au-CM concentration. Ag-CM induced cellular toxicity on DLD-1 at 5 μg/mL, a phenomenon amplified by increasing Ag-CM concentration. Maximum toxicity was observed when cells were exposed to a concentration of 100 μg/mL Ag-CM (Figure 7).

Based on the viability assay, the concentrations chosen for further experiments were 40 μg/mL CM extract, 40 μg/mL Au-CM, and 4 μg/mL Ag-CM. These doses caused significant toxicity to DLD-1 but were not toxic to normal cells (HDF).

#### 3.2.2. Cell Death Mechanism

The mechanism of cell death was assessed on DLD-1 cells by flow cytometry using annexin-V/propidium iodide staining. Exposure of DLD-1 cells to CM extract reduced cell viability (*p* < 0.05); the main mechanism involved necrosis. Au-CM did not significantly alter cell viability compared to control cells, while administration of Ag-CM caused a significant decrease in cell viability (*p* < 0.01), predominantly by apoptosis (*p* < 0.001) (Figure 8).

The protein expressions of p53, BAX, BCL-2, NF-κB and pNF-κB were quantified by Western blot analysis (Figure 9). p53 protein increased significantly after nanoparticle exposure compared to control cells (*p* < 0.001 for Au-CM and *p* < 0.01 for Ag-CM). BAX protein was significantly enhanced only when CM extract was administered (*p* < 0.05), while BCL-2 significantly decreased after administration of all treatments (*p* < 0.001), with a minimal value for Au-CM (Figure 9).

The nuclear factor NFkB is a transcription factor involved in immune and inflammatory responses. It is involved in cell proliferation and apoptosis. NFkB increased significantly (*p* < 0.001) after exposure to Au-CM and Ag-CM and decreased (*p* < 0.01) after incubation with CM extract. The phosphorylated form of NFkB was significantly enhanced (*p* < 0.001) after CM treatment and decreased significantly (*p* < 0.001) after Ag-CM administration (Figure 9).

The nuclear proteins Ki-67 and PCNA (proliferating cell nuclear antigen) are specific markers of cell division. The nuclear protein Ki-67 is increased in proliferating cells, but is suppressed in quiescent cells (G0 phase) [Schluter et al., 1993]. Ki-67 protein significantly decreased after nanoparticle exposure compared to untreated cells (*p* < 0.001 for Au-CM and *p* < 0.01 for Ag-CM). PCNA protein significantly diminished after nanoparticle administration compared to control cells (*p* < 0.05 for Au-CM and *p* < 0.001 for Ag-CM) (Figure 10). Mitogen-activated protein kinases (MAPKs) are a family of protein kinases involved in proliferation, differentiation and cell death. The expression of MAPK 42/44 (*p* < 0.001) decreased after Ag-CM administration compared to control, untreated cells (Figure 10). 

Caspases are enzymes involved in apoptosis. Caspase 3 showed no significant changes after exposure to CM extract or Au-CM, but decreased significantly (*p* < 0.01) in the presence of Ag-CM treatment. Caspase 8 levels were significantly increased after administration of all treatments compared to control cells (*p* < 0.05 for CM extract, *p* < 0.001 for Au-CM and *p* < 0.01 for Ag-CM) (Figure 11).

#### 3.2.3. Oxidative Stress Assessment

Malondialdehyde (MDA) is a specific marker of lipid peroxidation. This increased significantly after nanoparticle treatment (*p* < 0.01 for Au-CM and *p* < 0.001 for Ag-CM), compared to control cells (Figure 12).

## 4. Discussion

Metallic nanoparticles are widely used in cancer therapy for the delivery of bioactive molecules or drugs to the target site. Nanoparticles obtained by phytoreduction have a superior antiproliferative activity compared to their chemically synthesized counterparts due to the presence of natural compounds, which can sometimes have an additive effect with nanoparticles [28].

In our research, the antiproliferative effect of Au-CM and Ag-CM nanoparticles coated with CM fruit extract on DLD-1 tumor cell line was evaluated. The concentrations used, chosen on the basis of the viability assay, were not toxic to normal cells, but caused significant decreases in the viability of DLD-1 cells, suggesting cytotoxic effects on tumor cells.

The nanoparticles were obtained by green synthesis using CM extract as a source of reducing agents. CM extract was chosen due to its cytotoxic effect demonstrated on melanoma tumor cell lines (A375 and MeWo) [42], colon adenocarcinoma cells (Caco-2) [43] or on gastric carcinoma cells (AGS) [44]. Other studies have shown moderate antiproliferative effects of this extract on human tumor cells (breast adenocarcinoma—MCF-7, hepatocellular carcinoma—HepG2, colon adenocarcinoma—Caco2, HT-29) or murine tumor cells (colon carcinoma—CT26) [45].

In our study, FACS analysis showed that CM extract reduced cell viability by 93.3%, similar to that observed after Au-CM treatment (92.8%). In contrast, exposure of cells to Ag-CM significantly decreased cell viability (75.9%) compared to 99.4% for control, unexposed cells. Au-CM administration induced apoptosis (5.3%) and necrosis (1.9%) as cell death mechanisms, while Ag-CM was largely proapoptotic (21.4%). These results demonstrated the greatest toxicity of Ag-CM on tumor cells through apoptosis.

Western blot analysis revealed increased proapoptotic p53 protein and decreased BCL2 expression after exposure of tumor cells to both types of nanoparticles. These results suggested apoptosis as the main mechanism of cell death, and they are consistent with viability analysis and FACS analysis.

In general, apoptosis can be triggered extrinsically, by activating the Fas receptor, or intrinsically, a process controlled by BCL2 family proteins. Signals from the Fas receptor induce the formation of a complex consisting of FADD (Fas Associated Death Domain) and procaspase-8, which will cleave caspase-8. The apoptotic signal propagates either through cleavage and direct activation of downstream caspases or through cleavage of the BH3-interacting protein BCL2. The final result is the release of cytochrome c from the mitochondria [46]. In our study, nanoparticle administration significantly increased caspase 8 levels, which also indicates apoptosis as a mechanism of tumor cell death.

In our study, nanoparticle administration significantly increased caspase 8 levels, which also indicates apoptosis as a mechanism of tumor cell death.

Cell proliferation indices (nuclear proteins Ki-67 and PCNA) significantly decreased after administration of both types of nanoparticles, indicating the arrest of cell division. Ki67 is frequently used in oncology as a proliferation indicator. Ki-67 has a high level of expression during cell division and shows an increased risk of tumor invasion and metastasis [47]. Studies in the literature have shown a reduced survival in colorectal cancer patients when they had an increased Ki-67 expression [48,49,50].

PCNA can be used as a biomarker of colorectal adenocarcinoma [51]. It has been shown that an increased value of PCNA correlates with a high degree of malignancy and metastasis and a reduced degree of survival.

The p44/42 MAPK (Erk1/2) signaling pathway can be activated in response to a diverse range of extracellular stimuli, including mitogens, growth factors, and cytokines [52,53,54]. It is used as a target in the diagnosis and treatment of cancer [55]. In our study, all these parameters decreased significantly only after the administration of silver nanoparticles, suggesting the potent antitumor property of AgCM.

The different biological responses of tumor cells to the administration of nanoparticles depend on their physicochemical properties (shape, size, electrical charge and/or surface molecules), but also on the type of cells or tissue microenvironment [56].

The size of nanoparticles is recognized as an important factor for their uptake into cells [57]. The literature studies have shown that spherical and small-sized AuNPs are internalized into cells more efficiently compared to other shapes. Cho et al. (2010) exposed SK-BR-3 breast tumor cells to spherical or cubic gold nanoparticles of different sizes and functionalized with different substances. The results revealed a higher internalization for the spherical AuNPs compared to the cubic ones; for small compared to large AuNPs; for AuNPs functionalized with polyallylamine hydrochloride compared to those functionalized with anti-HER2 antibodies or polyethylene glycol [58]. Pramanik et al. (2022) used cubosomes functionalized with Affimer proteins and demonstrated, in in vitro and in vivo studies, their preferential accumulation in tumor cells and low toxicity on normal cells (biblio cerută cubozomi) [59].

In our study, XRD analysis confirmed the formation of silver and gold nanoparticles. TEM images showed that the nanoparticles used were spherical in shape and had an average diameter of 17 nm for Ag-CM and 20 nm for Au-CM. The hydrodynamic diameter of the nanoparticles was 326 nm for Ag-CM and 229 nm for Au-CM, respectively, demonstrating the binding of polyphenols to the surface of the nanoparticles. The polydispersity index (PDI) was high, suggesting a non-uniform distribution of metallic nanoparticles. Kovács et al. (2016) observed on osteosarcoma cells that silver nanoparticles, once internalized, trigger identical apoptotic pathways, regardless of their size [60].

The zeta potential was −31 mV for Ag-CM and −24.4 mV for Au-CM. These values demonstrated that the particles used were stable and had a low aggregation capacity. In addition, the negative electrical charge, similar to that on the cell membrane, removes the nanoparticles from the cells, reducing their toxicity. It is known that the absorption and cellular internalization of nanoparticles is favored by the presence of electrical charges opposite to those on the surface of the cell membrane [61,62].

The cytotoxic activity of nanoparticles is mediated by increased ROS production, oxidative stress with irreversible cell damage, DNA damage and cell death by apoptosis, autophagy or necrosis [63,64,65,66,67].

Mitochondria are the main target of ROS [68]. Nanoparticles cause damage to the electron transport chain, structural damage, activation of NADPH enzymes, and mitochondrial membrane depolarization [69,70,71,72,73]. Consequently, mitochondrial membrane permeability increases with the release of molecules such as cytochrome-c, apoptosis-inducing factor, and endonuclease G [74,75]. Cytochrome-c released into the cytosol causes the activation of caspase-9 [76], which cleaves and activates caspases 3 and 7, which can cause cell death. In addition, apoptosis-inducing factor and endonuclease G cause caspase-independent apoptosis [74,77].

In our study, nanoparticles induced oxidative stress, increased p53 protein expression and decreased BCL2 in DLD-1 cells. All these changes are indicators of the presence of apoptosis. However, cell viability decreased significantly only after exposure to Ag-CM, while Au-CM did not induce substantial changes in cell viability.

In the literature, other authors also reported the lack of cytotoxic effects of gold nanoparticles. Egbuna et al. (2021) showed that chemically synthesized gold nanoparticles with sizes below 20 nm were toxic to stem cells, but those obtained by green synthesis proved to be effective carriers of anticancer drugs with minimal side effects. In fact, the toxicity of gold nanoparticles depends on the synthesis method, size, morphology and surface functional groups [78].

In addition, in a study on human leukemia K562 cells, Connor et al. (2005) demonstrated that gold nanoparticles of different sizes and different capping agents penetrate tumor cells, but do not induce toxicity [79]. Similar results were observed by other researchers on Raw264.7 mouse macrophages [80] or on dendritic cells [81].

Nanoparticles obtained by phytosynthesis carry biologically active compounds from the natural extract, which can modulate their antitumor activity. In addition, tumor cell viability after nanoparticle exposure also depends on tumor characteristics. In a previous study, Baldea et al. (2020) demonstrated cytotoxic effects on dysplastic oral keratinocytes (DOK) after exposure to both gold and silver nanoparticles phytosynthesized with *Cornus mas* extract [41]. Silver nanoparticles capped with compounds from *Sambucus nigra* extract induced ultrastructural changes in DOK and inhibited pro-survival molecules and apoptosis regulators, decreased oxidative stress and inflammation, and induced cell death through necrosis, autophagy, and DNA damage [82]. All these data provide arguments for the potential antitumor role of silver nanoparticles without significant side effects on normal cells, especially on some types of tumors, and prove their utility in the treatment of various biomedical conditions.

## 5. Conclusions

Both gold and silver nanoparticles obtained by phytoreduction with bioactive compounds from *Cornus mas* extract protected normal cells and induced cell death in DLD-1 cells, especially by apoptosis.

However, further studies are needed to evaluate the effects of long-term exposure to phytosynthesized nanoparticles so as to minimize the side effects and maximize the benefits of their use in antitumor treatments.

## Figures and Tables

**Figure 1 nanomaterials-13-01251-f001:**
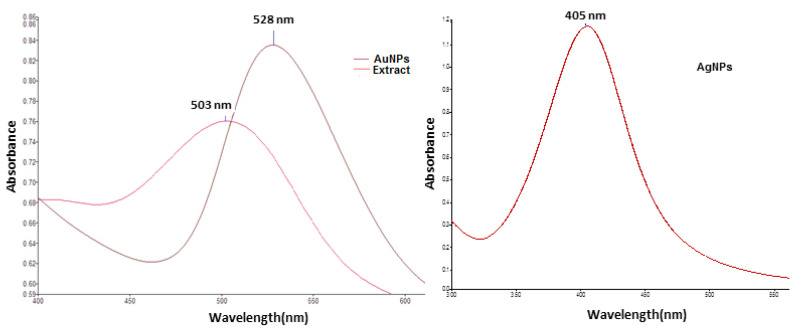
UV–Vis spectra of fruit extract, silver and gold nanoparticles biosynthesized with *Cornus mas* fruit extract as source of reducing and capping agents.

**Figure 2 nanomaterials-13-01251-f002:**
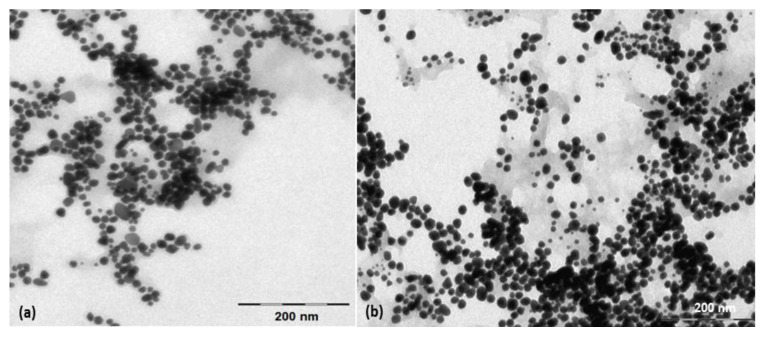
TEM (transmission electron microscopy) images of Ag-CM nanoparticles (**a**) and Au-CM nanoparticles (**b**) biosynthesized with *Cornus mas* fruit extract as source of reducing and capping agents.

**Figure 3 nanomaterials-13-01251-f003:**
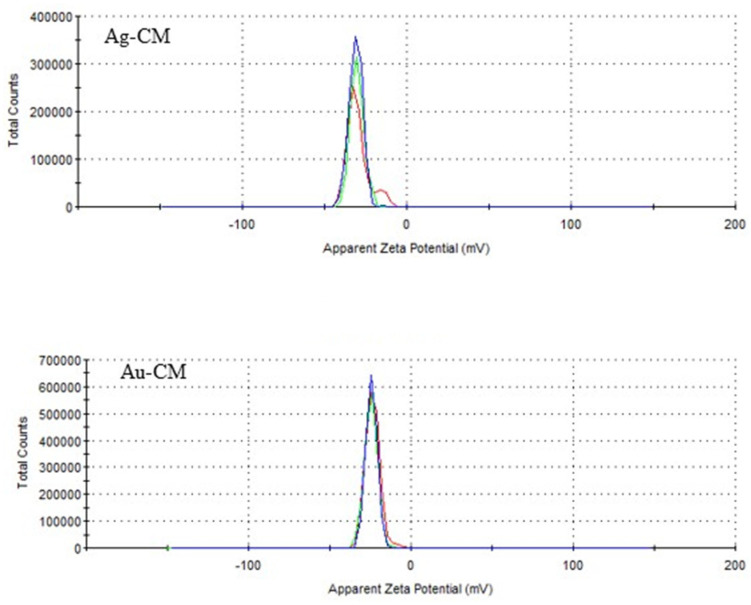
Zeta potential of silver and gold nanoparticles biosynthesized with *Cornus mas* fruit extract as source of reducing and capping agents.

**Figure 4 nanomaterials-13-01251-f004:**
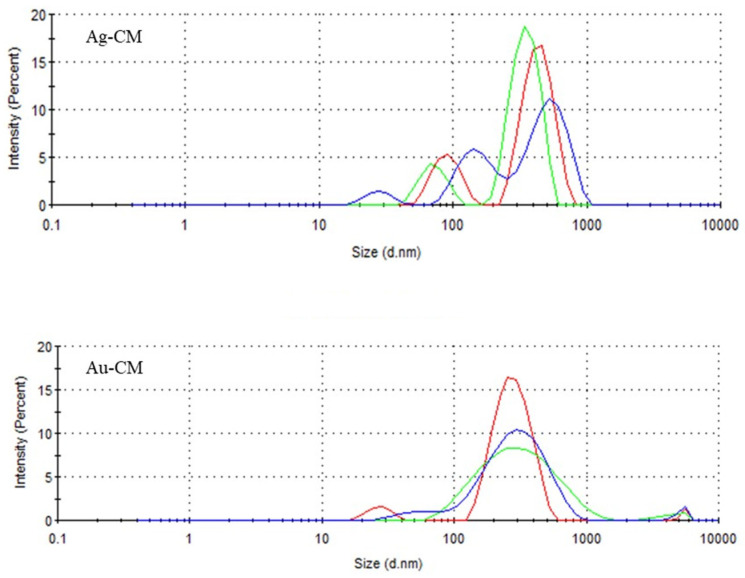
The hydrodynamic diameter of the silver and gold nanoparticles biosynthesized with *Cornus mas* fruit extract as source of reducing and capping agents.

**Figure 5 nanomaterials-13-01251-f005:**
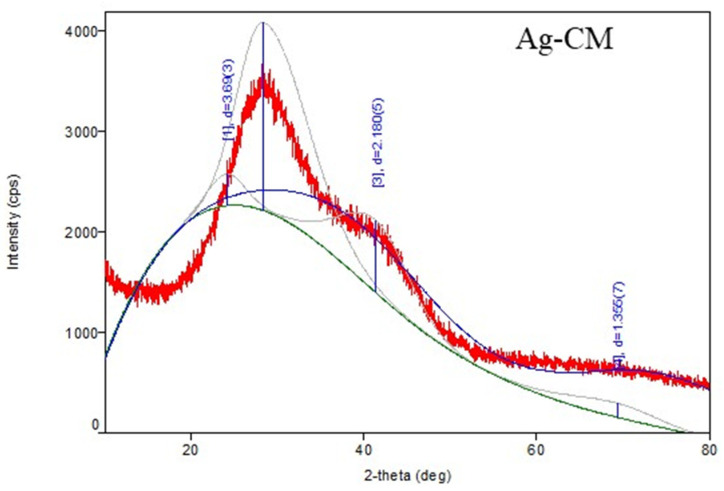
XRD (X-ray powder diffraction) analysis for silver and gold nanoparticles biosynthesized with *Cornus mas* fruit extract as source of reducing and capping agents.

**Figure 6 nanomaterials-13-01251-f006:**
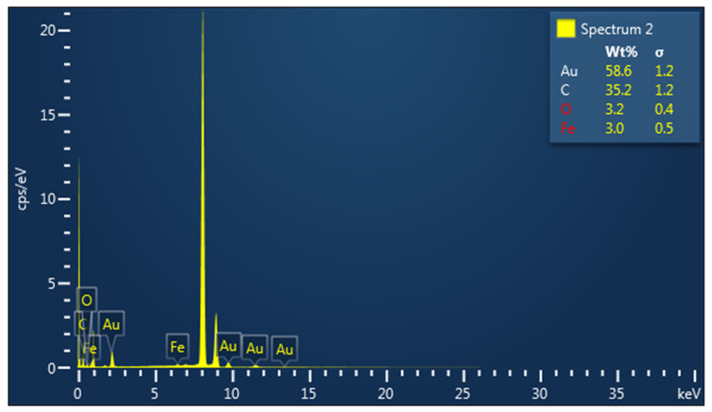
EDX (energy dispersive X-ray diffraction) analysis for silver and gold nanoparticles biosynthesized with *Cornus mas* fruit extract as source of reducing and capping agents.

**Figure 7 nanomaterials-13-01251-f007:**
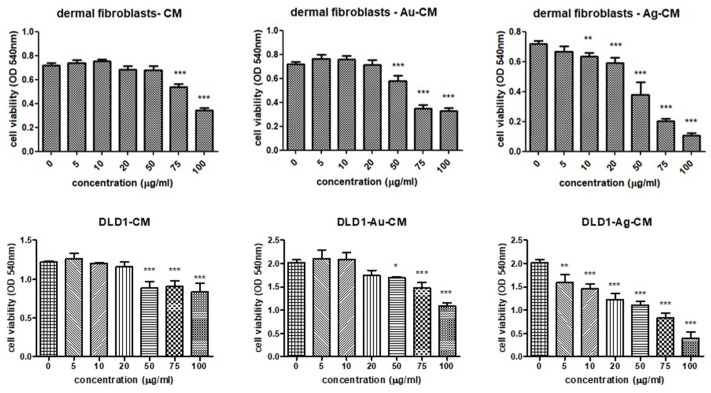
Viability of HDF and DLD-1 cells exposed to Au-CM and Ag-CM at different concentrations (0–100 μg/mL) compared to unexposed control cells or cells exposed to CM extract. Data are presented as mean OD 540 nm ± SD, (n = 3) (* *p* < 0.05, ** *p* < 0.01, *** *p* < 0.001 treated cells vs. untreated control cells).

**Figure 8 nanomaterials-13-01251-f008:**
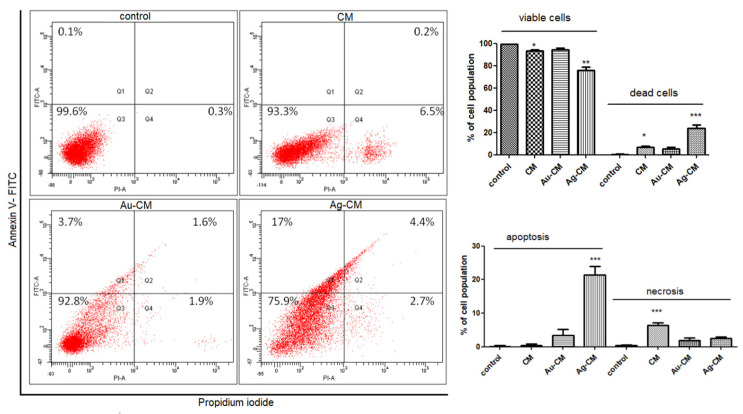
Flow cytometric analysis of DLD-1 cells treated with CM extract, Au-CM or Ag-CM. Staining was performed with annexin V/PI. CM extract decreased cell viability by necrosis, while Ag-CM significantly decreased cell viability mainly by apoptosis. Quantitative FACS results for DLD-1 are expressed as % of total cell number (left panel). The quadrants have the following meaning: viable cells (bottom left), early apoptosis (top left), late apoptosis (top right), necrosis (bottom right). Statistical analysis was performed with one-way ANOVA and Bonferroni’s posttest (* *p* < 0.05, ** *p* < 0.01, *** *p* < 0.001 treated cells vs. untreated control cells).

**Figure 9 nanomaterials-13-01251-f009:**
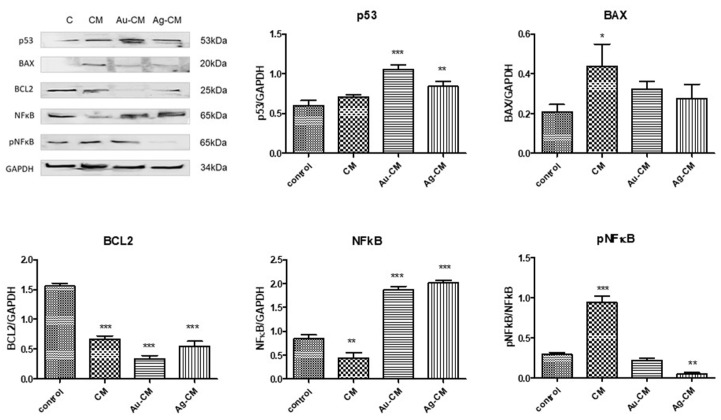
The protein expressions of p53, BAX, BCL2, NFkB and pNFkB in DLD-1 cells by Western blot examination. DLD-1 cells were treated with 40 μg/mL CM extract, 40 μg/mL Au-CM, and 4 μg/mL Ag-CM. In the upper panel are the images of the Western blot bands; in the lower panel are the results normalized to GAPDH. Statistical analysis was performed with one-way ANOVA and Bonferroni’s posttest (* *p* < 0.05, ** *p* < 0.01, *** *p* < 0.001 compared to control cells).

**Figure 10 nanomaterials-13-01251-f010:**
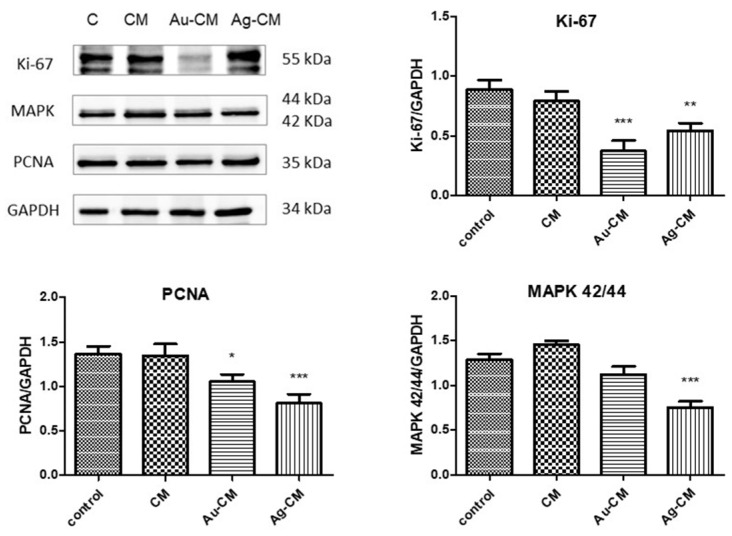
Ki-67, PCNA and MAPK 42/44 expression in DLD-1 cells evaluated by Western blot. DLD-1 cells were treated with 40 μg/mL CM extract, 40 μg/mL Au-CM, and 4 μg/mL Ag-CM. In the upper panel are the images of the Western blot bands; in the lower panel are the results normalized to GAPDH. Statistical analysis was performed with one-way ANOVA and Bonferroni’s posttest (* *p* < 0.05, ** *p* < 0.01, *** *p* < 0.001 compared to control cells).

**Figure 11 nanomaterials-13-01251-f011:**
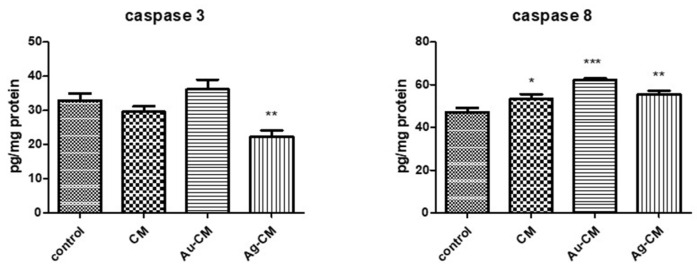
Caspase 3 and 8 in DLD-1 cells after treatment with CM extract, Au-CM and Ag-CM. DLD-1 cells were treated with 40 μg/mL CM extract, 40 μg/mL Au-CM, and 4 μg/mL Ag-CM for 24 h and then the levels of caspase 3 and 8 were measured by using ELISA kit. Statistical analysis was performed with one-way ANOVA and Bonferroni’s posttest (* *p* < 0.05, ** *p* < 0.01, *** *p* < 0.001 compared to control cells).

**Figure 12 nanomaterials-13-01251-f012:**
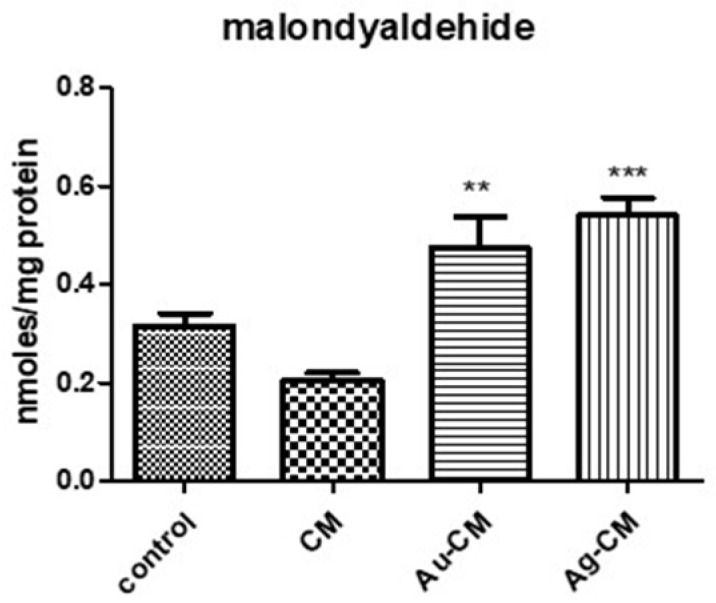
Malondyaldehide in DLD-1 cells after treatment with CM extract, Au-CM and Ag-CM. DLD-1 cells were treated with 40 μg/mL CM extract, 40 μg/mL Au-CM, and 4 μg/mL Ag-CM for 24 h, and then MDA levels were assessed using the ELISA kit. Statistical analysis was performed with one-way ANOVA and Bonferroni’s posttest (** *p* < 0.01, *** *p* < 0.001 compared to control cells).

## Data Availability

Not applicable.

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
