# Peer review of "Green Synthesized Gold and Silver Nanoparticles Increased Oxidative Stress and Induced Cell Death in Colorectal Adenocarcinoma Cells"

_nanomaterials, 2023, doi:10.3390/nano13071251_

Round 1
Reviewer 1 Report
The author here used phytoreduced gold (Au-CM) and silver nanoparticles (Ag-CM) with Cornus mas (CM) fruit to inhibit colorectal adenocarcinoma cell lines (DLD-1) in vitro. The prepared nanoparticles were characterized UV spectroscopy, transmission electron microscopy, potential zeta and hydrodynamic diameter measurements, energy dispersive X-ray and X-ray powder diffraction analysis before biological research. The nanoparticles induced apoptosis and necrosis of DLD-1 cells, and maintained up to 80% viability. Ag-CM and Au-CM up-regulated p53 and NF-kB level in parallel with the downregulation of BCL-2, and induced the activation of caspase-8. Lipid peroxidation triggered by Ag-CM was correlated with tumor cell necrosis rate. Basically, their data are interesting for cancer therapy. However, there are several concerns before publication.
Major points:
1. Only in vitro data was collected in the current study, which will decrease the significance of conclusion.
2. Biological analysis was not enough. Only apoptosis, viability and some proteins were analyzed. The cell cycle, proliferation and tumor growth were not taken into consideration, which imposed uncertain outcome after these nanoparticles administration.
Minor points:
1. The figure legend is too simple to be understood, especially from Fig.1 to Fig.6.
2. In Fig.9, the Western blot data should be marked in each lane.
3. What’s the main components of cornelian cherries extract? Could the author provide any major chemical structure of this extract? It is a single compound or chemical mixture?
Author Response
Reviewer 1
Major points:
- Only in vitrodata was collected in the current study, which will decrease the significance of conclusion.
Response: The current manuscript was focused on in vitro data on DLD cells. Another study on in vivo carcinogenesis is being prepared for publication.
- Biological analysis was not enough. Only apoptosis, viability and some proteins were analyzed. The cell cycle, proliferation and tumor growth were not taken into consideration, which imposed uncertain outcome after these nanoparticles administration.
Response: The proliferation markers (Ki 67, PCNA and MAPK) were evaluated additionally in tumor cells as the you suggested.
Material and Methods
The nuclear proteins Ki-67 and PCNA (proliferating cell nuclear antigen) are specific markers of cell division. The nuclear protein Ki-67 is increased in proliferating cells, but is suppressed in quiescent cells (G0 phase) [Schluter et al. 1993]. Ki-67 protein significantly decreased after nanoparticle exposure compared to untreated cells (p<0.001 for Au-CM and p<0.01 for Ag-CM). PCNA protein significantly diminished after nanoparticle administration compared to control cells (p<0.05 for Au-CM and p<0.001 for Ag-CM) (Figure 10). Mitogen-activated protein kinases (MAPKs) are a family of protein kinases involved in proliferation, differentiation and cell death. The expression of MAPK 42/44 (p<0.001) decreased after Ag-CM administration compared to control, untreated cells (Figure 10).
Figure 10. Ki 67, PCNA and MAPK 42/44 expression in DLD-1 cells evaluated by western blot. DLD-1 cells were treated with 40 μg/ml CM extract, 40 μg/ml Au-CM, and 4 μg/ml Ag-CM. In the upper panel are the images of the western blot bands; in the lower panel are the results normalized to GAPDH. Statistical analysis was performed with one-way ANOVA and Bonferroni's posttest (*p<0.05, **p<0.01, ***p<0.001 compared to control cells).
Discussion
...
Cell proliferation indices (nuclear proteins Ki-67 and PCNA) significantly decreased after administration of both types of nanoparticles, indicating the arrest of cell division. Ki67 is frequently used in oncology as a proliferation indicator. Ki-67 has a high level of expression during cell division and shows an increased risk of tumor invasion and metastasis (54). Studies in the literature have shown a reduced survival in colorectal cancer patients when they had an increased Ki-67 expression (Salminen et al. [8], Xi et al. [15], Ivanecz et al. [16]).
PCNA can be used as a biomarker of colorectal adenocarcinoma (6). It has been shown that an increased value of PCNA correlates with a high degree of malignancy and metastasis and a reduced degree of survival.
The p44/42 MAPK (Erk1/2) signaling pathway can be activated in response to a diverse range of extracellular stimuli, including mitogens, growth factors, and cytokines (Philippe P Roux 2004, Manuela Baccarini 2005, S Meloche 2007) . It is used as a target in the diagnosis and treatment of cancer (P J Roberts 2007). In our study, all these parameters decreased significantly only after the administration of silver nanoparticles suggesting the potent antitumor property of AgCM.
Minor points:
- The figure legend is too simple to be understood, especially from Fig.1 to Fig.6.
Response: The figure legend was expanded in order to be better understood.
- In Fig.9, the Western blot data should be marked in each lane.
Response: The western blot data was marked in each lane.
- What’s the main components of cornelian cherries extract? Could the author provide any major chemical structure of this extract? It is a single compound or chemical mixture?
Response: A new paragraph was added to the Introduction section of the manuscript, explaining which are the main components of the investigated fruit extract as determined in one of our previous studies.

Reviewer 2 Report
The authors have performed a green synthesis of gold and silver nanoparticles an studied the anti-tumor activity of these NPs. The work is indeed very interesting.
There are certain queries for the authors to address.
How could the selective release of Ag ions at tumor site could be achieved for clinical translation of the NPs.
What was the polydispersity index of the nanoparticles (silver and gold nanoparticles). It is observed that the z-average peak (DLS) from figure 4, is quite wide which could mean the PDI could be high.
Metal based nanoparticles of copper have been widely studied for colorectal cancers and authors could include this in their discussion (https://doi.org/10.1021/acsami.1c21655). Similarly metal nanoparticles have also been used for targeted delivery in colorectal cancer spheroid models such as - using Hyaluronic-Acid-Tagged Cubosomes to deliver cytotoxics specifically to CD44-Positive Cancer Cells.
Author Response
Reviewer 2
The authors have performed a green synthesis of gold and silver nanoparticles an studied the anti-tumor activity of these NPs. The work is indeed very interesting.
There are certain queries for the authors to address.
- How could the selective release of Ag ions at tumor site could be achieved for clinical translation of the NPs.
Response: AgNPs enter cells by endocytosis and are transferred to lysosomes. The acidic lysosomal environment degrades the nanoparticles with the release of Ag+ ions into the cytosol [Valeria De Matteis et al. 2015]. Once internalized, nanoparticles release high levels of toxic ions (“Trojan horse” mechanism) [I-Lun Hsiao et al. 2015].
What was the polydispersity index of the nanoparticles (silver and gold nanoparticles). It is observed that the z-average peak (DLS) from figure 4, is quite wide which could mean the PDI could be high.
Response: The PDI for AuCM is 9.457 and for AgCM is 0.481 suggesting a not uniform distribution of metallic nanoparticles.
Metal based nanoparticles of copper have been widely studied for colorectal cancers and authors could include this in their discussion (https://doi.org/10.1021/acsami.1c21655). Similarly metal nanoparticles have also been used for targeted delivery in colorectal cancer spheroid models such as - using Hyaluronic-Acid-Tagged Cubosomes to deliver cytotoxics specifically to CD44-Positive Cancer Cells.
Response: The new bibliography was added as following: Pramanik et al. (2022) used cubosomes functionalized with Affimer proteins and demonstrated, in in vitro and in vivo studies, their preferential accumulation in tumor cells and low toxicity on normal cells ().
Round 2
Reviewer 1 Report
All the concerns were addressed in the current edition.